# Vitamin D and Male Reproduction: Updated Evidence Based on Literature Review

**DOI:** 10.3390/nu14163278

**Published:** 2022-08-10

**Authors:** Gloria Calagna, Valeria Catinella, Salvatore Polito, Antonio Schiattarella, Pasquale De Franciscis, Francesco D’Antonio, Giuseppe Calì, Antonino Perino, Gaspare Cucinella

**Affiliations:** 1Obstetrics and Gynecology, “Villa Sofia Cervello” Hospital, IVF Unit, University of Palermo, Via Trabucco 180, 90145 Palermo, Italy; 2Obstetrics and Gynecology, “Fondazione per gli Studi sulla Riproduzione Umana”–Clinica Candela, Via Valerio Villareale 54, 90141 Palermo, Italy; 3Department of Woman, Child and General and Specialized Surgery University of Campania “Luigi Vanvitelli”, Largo Madonna delle Grazie, 1, 80138 Naples, Italy; 4Center for High-Risk Pregnancy and Fetal Care, Department of Obstetrics and Gynecology, University of Chieti, Via dei Vestini 31, 66100 Chieti, Italy; 5IVF Unit, Azienda Sanitaria Provinciale di Palermo, Via G. Cusmano 24, 90100 Palermo, Italy

**Keywords:** vitamin D, infertility, male reproduction, semen quality, sex hormone production, supplementation

## Abstract

The purpose of this study was to clarify the vitamin D (VD) effect on male infertility. Our research was conducted using the following electronic databases: MEDLINE, Embase, Web of Science, Scopus, ClinicalTrials.gov, and Cochrane Library. Selection criteria included all published randomized controlled trials and non-randomized studies, focusing on vitamin D and male reproductive function. The results showed that the effects of VD on male reproduction has been investigated in three different topics: the molecular mechanism underlying VD effects on semen quality (SQ), the relationship between VD levels and SQ, and the effect of VD supplementation on SQ. Results supported the hypothesis of a relevant interrelation between concentrations of male VD and semen parameters, with particular reference to sperm motility; on the contrary, evidence on the effect of VD on male sex steroid hormone levels was inconclusive. The results of this review hold up the thesis that VD plays a role in male reproduction. Most of the data highlighted a positive effect on semen quality, particularly in sperm motility, both in fertile and infertile men. Additional dedicated studies are required to elucidate the still controversial aspects of this topic.

## 1. Introduction

Vitamin D (VD) is a group of liposoluble vitamins, synthesized in the skin due to Ultraviolet B (UVB) sun radiation; it has the characteristic to bind to a classical steroid hormone receptor, such as it was a “pro-hormone” [1,2]. VD plays a key role in the regulation of calcium metabolism, phosphate homeostasis and bone health. The classical effects of VD are expressed in the sites where VD receptor (VDR) activation promotes trans-cellular calcium and phosphate absorption and reabsorption, like the intestine and kidney [2,3,4]. VD deficiency can be the result of genetic defects, inadequate dietary intake and/or insufficient cutaneous production [5,6,7].

More recently, VD has been defined as a “pleiotropic” molecule with several hypothesized actions and functions [8,9]. In the human reproductive field, the documented high expression of VDR with VD-metabolizing enzymes in the male reproductive system was associated to the possible role of VD synthesis and regulation, as well as function in the testis [10]. In particular, VDR and VD-metabolizing enzymes were concomitantly expressed in spermatids, vesicles seminal vesicles and prostate, suggesting that VD could be implicated in spermatogenesis and maturation of human spermatozoa [10,11,12]. Moreover, VD has been hypothesized to be an important factor for trans-epithelial calcium transfer in the epididymis [10]. All these data results encouraged the idea that expression of VD was a possible marker of semen quality (SQ) [12].

Recent in vivo data confirmed the association between VD, reproductive hormones and sperm parameters in infertile males, supporting the hypothesis that VD could have a positive effect on male reproductive capacity [13,14]. Further in-depth studies showed that men with VD deficiency or insufficiency resulted in significantly lower sperm motility than men with normal VD levels [15]. Although a relationship between VD and SQ has often been proposed (with a respective potential role of VD in the fertilization process), actual literature data on the topic was controversial and conflicting [16,17,18,19].

To better clarify the main aspects of the relationship between VD and male reproduction, we performed a systematic review of the inherent literature with particular attention to the effects of VD on male infertility.

## 2. Materials and Methods

The research was conducted using the following electronic databases: MEDLINE, Embase, Web of Science, Scopus, ClinicalTrials.gov, and Cochrane Library. The studies were identified using combinations of the search terms “male reproduction”, “sperm”, “testis”, “gonad”, “fertility”, “epididymis”, “semen quality”, “sex hormones”, “testosterone”, in combination with “vitamin D”, “vitamin D levels”, “vitamin D receptor”, “vitamin D role”. We decided to limit the search to articles published from January 2010 to December 2020, in order to guarantee the relevance and up-to-dateness of the paper.

Only scientific publications in English were included. Selection criteria included all published randomized controlled trials (RCTs) and non-randomized studies (NRSs) (e.g., observational, prospective, retrospective cohort studies, case-control studies), focusing on vitamin D and male reproductive function. Two independent reviewers (G.C. and V.C.) conducted the review steps (the electronic search, study eligibility, inclusion criteria, risk of bias, data extraction and analysis). Disagreements were resolved by consensus with a third reviewer (A.P.). All cross-references were also hand-searched. Proceedings of scientific meetings and abstracts were not considered. We excluded from the analysis the experimental in vitro studies or on animal models.

All articles describing the effects of VD on SQ and/or male fertility were considered. Only original papers that reported specific experience data on the topic were included. All the relevant aspects of the included papers were recorded and commented. The review was reported following the Preferred Reporting Item for Systematic Reviews and Meta-Analyses (PRISMA) statement [20]. Local Scientific Committee approved the study. We applied the recent levels of evidence published by the Center for Evidence-Based Medicine in 2009 to describe the strength and the level of evidence of the results [21].

## 3. Results

We found 1283 records. Twenty-seven articles were selected as possibly relevant. Ten studies were then excluded as they were review or comment articles not reporting original data (*n* = 6) or animal studies (*n* = 4). Finally, 17 studies were included for the analysis [10,12,13,14,15,16,18,22,23,24,25,26,27,28,29,30,31] (Figure 1). The level of evidence of the achieved results are described in Table 1.

The results of our analysis showed that the effects of VD on SQ and/or fertility has been investigated in three types of evaluations: a. the molecular mechanism underlying VD effects on SQ (on human spermatozoa); b. the relationship between VD status and SQ; c. the effect of VD supplementation on SQ.

Based on these considerations, we divided the achieved included articles into sections, based on the specific topic:VD molecular mechanism [10,12,27];VD level and SQ [13,14,15,16,23,24,25,26,29,30,31];VD supplementation and SQ [5,22,28].

### 3.1. VD Molecular Mechanism

Poor data exist on the VD metabolizing enzymes (CYP2R1, CYP27A1, CYP27B1 and CYP24A1) expressed in human testis and the male reproductive tract. In 2010, Blomberg Jensen et al. demonstrated the expression of VDR and the VD metabolization in human testis and epithelia of the ejaculatory tract. Based on tissue samples obtained from orchiectomy (testis *n* = 13; epididymis *n* = 7), prostatectomy (prostate *n* = 5 and vesicles *n* = 3) and semen samples (*n* = 13), the authors found expression of VDR and VD metabolizing enzymes in germ cells during spermatogenesis, persisting in the ejaculatory duct and in ejaculated spermatozoa [10]. In particular, VDR mRNA was detected in testis, epididymis, prostate, seminal vesicles, and ejaculated spermatozoa, with variable levels of expression; the predominant spermatozoa sites of expression were the post-acrosomal region, neck and in the mid-piece of spermatozoa. On the basis of these results, the authors suggested an important role of VD for spermatogenesis and maturation of human spermatozoa [10].

Successively, in 2012, the same group of authors investigated the CYP24A1 enzyme expression as a possible marker of VD metabolism in spermatozoa and its association with SQ [12]. The VD-inactivating enzyme CYP24A1 regulates the responsiveness to VD by the cells and it is transcriptionally regulated by VD. Based on their study on semen analysis and immunocytochemical detection of the enzyme of 130 men (53 healthy and 77 sub-fertile men), the authors concluded that sperm enzyme expression in sub-fertile and fertile men (1% vs. 25%) had a positive correlation with total sperm count (45 vs. 192 × 106/mL), concentration (10 vs. 52 × 106/mL), motility (52% vs. 70%), and morphology (2.5% vs. 7.0%), with *p* < 0.0005. They also observed that the presence of >3% enzyme-positive spermatozoa distinguished young healthy men from sub-fertile men (PPV of 98.3%) [12].

Recently, Jueraitetibaike et al. explored sperm kinetic parameters in vitro after incubation for 30 min with 1,25 (OH)2 VD, verifying an increased upward migration of spermatozoa with growing adenosine triphosphate (ATP) levels [27]. They noted high levels of cyclic adenosine monophosphate (cAMP) elevated rate of activity of protein kinase A (PKA) and in addiction the PKA inhibitor turned the increase of ATP generation; moreover, the concentrations of cytoplasmic calcium ions and nicotinamide adenine dinucleotide (NADH) were either improved, while mitochondrial calcium uniporter inhibitor did not invert the ATP production rise. So, 1,25 (OH)2 VD could improve the sperm motility by promoting the synthesis of ATP, both through the cAMP/PKA pathway and the growth of intracellular calcium ions [27].

### 3.2. VD Level and SQ

In the last years, the reports of pandemic VD deficiency brought on the extension of the spectrum of extra-skeletal research on the topic and in particular more interest has been focused on the relationship between VD and SQ, with attention also on related sex hormonal values. All the main findings of this review on this topic are described in Table 2.

In a cross-sectional association study on 300 men from the general population, Blomberg Jensen et al. in 2011, aimed to show the role of activated VD in human spermatozoa and the hypothesis of association between VD serum levels and SQ [29]. A positive association between VD serum levels with and progressive motility (*p* < 0.05) was noted; in the case of VD deficiency (<25 nM), males had a lower proportion of motile (*p* = 0.027), progressive motile (*p* = 0.035) and morphologically normal spermatozoa (*p* = 0.044) in comparison with with high VD levels males (>75 nM). The authors reported an additional in vitro result: via VDR-mediated calcium release from an intracellular calcium storage, VD increased intracellular calcium levels in human spermatozoa rise the sperm motility and produce the acrosome reaction [29]. In the same year, Ramlau-Hansen et al. published their data on the association between serum VD concentrations, SQ and levels of reproductive hormones in a population of 307 young adult men with normal semen [30]. Unexpectedly, a high VD level was related to lower median total sperm count and normal morphology sperm rate; in particular, men with high VD (*n* = 101) had approximately 31% lower total sperm count and 23% lower normal morphology percentage compared with low (*n* = 103) and medium (*n* = 103) VD levels [30]. However, these trends were not statistically significant.

In 2012, Hammoud et al. evaluated SQ parameters in 3 groups of healthy patients, with different serum VD concentrations: group 1 (*n* = 19), VD < 20 ng/mL; group 2 (*n* = 108), VD > 20 and <50 ng/mL; group 3 (*n* = 20), VD > 50 ng/mL [15]. The relation between VD and hormonal and semen parameters was corrected for age, BMI, season, alcohol intake and smoking. Focusing on differences between group 1 (VD insufficiency) and group 2 (VD normal), authors reported that total sperm count (85.1 vs. 178.6) and total progressive motile sperm count (45.3 vs. 98.8) were lower in group 1 compared to group 2 (*p* < 0.05). No statistical difference between the mean hormonal values in the different categories of VD was seen [15].

In a cross-sectional study, Yang et al. aimed to investigate testosterone (T), VD, and SQ and their relationships in an infertile Chinese male population [31]. Fertile men (*n* = 195), infertile men (*n* = 9) with osteoporosis risk factors (WR) and infertile men (355) without osteoporosis risk factors (WOR) were enrolled; WOR men were divided into oligo-astheno-teratospermic (OATN, *n* = 314) and non-obstructive azoospermic (NOA, *n* = 41) groups. No correlations between T and VD were found in the examined groups. Focusing on the comparison between fertile and WOR-OATN men, a positive relationship between VD values and sperm motility as well as morphology in both study groups was found; in particular, OATN men with sufficient VD had a significantly higher percentage of morphologically normal spermatozoa than males with deficient (*p* = 0.005) and insufficient VD (*p* = 0.034) [31].

The observational study of Tartagni et al. evaluated the negative influence of low male VD serum levels on the conception rate in couples attempting pregnancy [16]. Ninety enrolled couples were divided into two study groups: 36 couples have been attributed to group 1 with normal VD levels (≥30 ng/mL), and 54 couples have been attributed to group 2 with low VD levels (<30 ng/mL). No substantial difference was observed in the sperm concentration, sperm progressive motility or sperm morphology in samples from the subjects of group 1 compared with subjects of group 2. Unexpectedly, the pregnancy rate (per patient and per cycle) was significantly higher (*p* < 0.05) in couples with normal VD levels, as well as delivery rate (per patient *p* < 0.02, per cycle *p* < 0.01) [16].

In 2016, Abbasihormozi et al., compared VD and hormone levels in normospermic (*n* = 186) and oligoasthenoteratozoospermic (OAT) (*n* = 92) men. VD levels revealed no association between sperm parameters and hormone balance in normospermic population; nevertheless, sperm motility was positive correlated with VD categorized in OAT men (*p* < 0.05) [23].

In the case-control study of Zhu et al., analysis of serum inactive VD (VD-OH) value reported no significant difference between infertile (*n* = 186) and fertile (*n* = 79) patients. However, VD levels had a positive statistically significant association with progressive motility and sperm count (*p* < 0.05) [25].

A study comprising a baseline of 1189 infertile men, screened for the Copenhagen Bone-Gonadal (CBG) study, who underwent a physical examination, semen evaluation and blood analysis for serum VD and hormonal profiles [14]. Positive linear associations between VD and progressive sperm motility were identified 45 min (*p* =0.04) and 4 h (*p* < 0.0005) after ejaculation; on average, the numbers of motile or progressive motile sperm were inferior in VD deficient men (*p* < 0.05). No significant relationship between lower total testos and VD deficiency was found [14].

A retrospective study by Tirabassi et al. showed the interconnection between serum VD and SQ parameters based on a case-series study of andrological patients. Normal total sperm motility males (*n* = 23) had significantly higher VD levels (*p* < 0.001) as compared to impaired total sperm motility (*n* = 81) [24]. Authors explained these findings based on the recent molecular evidence of the role of VD in spermatogenesis and sperm maturation in humans and in rats [24]. Akhavizadegan et al. in a retrospective case-control study (116 fertile vs. 114 infertile men) carried out in an endemic area of VD deficiency, supported these data, showing a positive association between high VD level and all SQ parameters (*p* < 0.001) [26].

In a cross-sectional study, Rehman et al. found that VD serum level seems to have a positive correlation progressive motility [13]. The cohort of 127 men (cases) with VD deficiency (<25 nmol/L) showed a lower rate of motile (*p* = 0.027), progressive motile (*p* = 0.035) and morphologically normal spermatozoa (*p* = 0.044) compared with men with high VD levels (186 controls). The median value of serum VD was significantly greater in the control group compared to the case group (*p* < 0.001); moreover, serum T level had a significant positive association with VD whereas luteinizing hormone (LH) had a significant negative association (*p* < 0.001). Limiting analysis to infertile subjects, high values of VD were observed in infertile samples with normal SQ (62.15 ± 13.86 nmol/L) in comparison to infertile men with altered SQ (51.87 ± 16.77 nmol/L) [13].

### 3.3. VD Supplementation and SQ

The Copenhagen Bone-Gonadal Study (CBG) is a randomized clinical trial carried out by Blomberg-Jensen et al., in which 330 infertile men with VD insufficiency (≤50 nmol/L) randomly received placebo or a high initial oral dose of VD dissolved in oil (300.000 UI, single dose), followed by 1400 UI VD and 500 mg of calcium daily for 150 days, in order to compare the outcomes on SQ, reproductive hormones levels and live birth rate [12]. Authors demonstrated that the high dose VD supplement increased semen volume in the treatment group (*p* < 0.03), but at day 150, there were not statistically significant difference about median sperm count and sperm concentration given the two study arms at day 150 (*p* = 0.07) [18]. However, the spontaneous pregnancy rate was higher in couples where the men were in the treatment group (live birth rate 7.3 vs. 2.4%: conceived spontaneously with *p* < 0.07); interestingly, analysis of a sub-group of oligozoospermic men with VD implementation showed an increased possibility of a live birth compared with placebo [18].

In a pilot-study, Waud et al. evaluated if semen parameters could be improved with repletion of daily oral VD for 90 days in a male population with abnormal SQ (*n* = 52) [22]. The mean serum VD level was 23.6 ng/mL at initial visit, and, after oral daily supplementation, VD levels were 30.1 ng/mL (*p* = 0.047) and sperm motility increased from 22.2 to 27.8%. Moreover, oral VD supplementation improved sperm motility (5.6%), however, not in a statistically significant manner [22]. In 2020, Wadhwa et al. reported the results of a longitudinal observational study on VD supplementation in 60 infertile men (oligoasthenozoospermia) with VD deficiency (<30 ng/mL) [28]. After administration of VD and calcium supplementation (60,000 IU/week cholecalciferol and 500 mg/day calcium) semen and hormone parameters were evaluated at the end of 3 and 6 months. After 6 months of VD supplementation, authors documented a significant improvement of the mean sperm concentration and progressive sperm motility in infertile males (*p* < 0.001). Finally, the overall study pregnancy rate was 8.33% after VD supplementation; however, this data was not statistically significant (*p* = 0.24) [28].

## 4. Discussion

In the last decade, interest in VD has grown, mainly after the identification of several “non-classical” targets of action, including the reproductive organs. The established expression of VDR and VD-metabolizing enzyme in the ovary, uterus, placenta and also in male reproductive tract and human spermatozoa, suggested an important role of VD for reproductive physiology regulation [10,32,33,34]. In particular, the presence of VD metabolizing enzymes in all reproductive tracts suggested the importance of locally regulated VD metabolism and also the evidence of VDR expression in the testis supported a VD role in influencing male infertility by autocrine and paracrine action [8,12,35].

Although it is a very interesting issue, our review of the international literature highlighted the presence of only a small number of articles on the role of VD in reproductive adulthood life and the fact that it has only been the subject of research in recent times [36,37,38,39]. Moreover, the obtained conclusions are mainly the result of cross-sectional observational studies. All these aspects could explain the scant knowledge on this topic. However, many data which have emerged must be commented on.

First of all, there are still few data on the mechanism of action of VD with reference to the male genital system and therefore several obscure points persist to this day. Starting from the evidence that the human testis and the ejaculatory tract are sites of metabolizing VD [10], the idea of an important role of local activation of VD in spermatogenesis and sperm maturation has grown in strength. In addition, CYP24A1 expression at the annulus of human spermatozoa was positively correlated with total sperm number, sperm concentration, sperm motility and morphology, being a novel marker for SQ [12]. Finally, despite the paucity of data, one aspect that can be agreed upon is the positive association between seminal plasma VD and sperm kinetic parameters: the activated VD may improve sperm motility through the promotion of the activation of the sperm mitochondrial respiratory chain to produce ATP and the increase of intracellular calcium ion concentration. In this sense, seminal plasma VD better reflects the status of male reproduction if compared with serum VD.

Certainly, the main object of our search was the correlation between male VD deficiency or insufficiency and characteristics of SQ. Although the correlation between VD, semen total count and morphology rate remain ambiguous, the results on association with sperm motility appear more significant. Most of the specifically related articles included in the review (9 out of 11 included articles) reported a positive association of VD serum level with semen total motility, both in fertile and infertile males; moreover, in many cases, this positive association of VD level and semen total motility had statistical significance [13,14,15,23,24,25,26,29,31]. On the contrary, evidence of the effect of VD on male sex steroid hormone production was less encouraging. Based on the results of the considered studies on the possible association between follicle-stimulating hormone, LH and testosterone, no relevant correlation was highlighted [13,14,15,23,24,29,30,31]. All these data suggested a favorable outcome of VD on male reproductive health, above all through the relation with improved sperm motility but also an inconsistent association with sex hormone production.

Another very interesting aspect of our research concerns the possible role of VD supplementation on SQ. Although overall results on the topic suggested that VD supplementation had no significative association with changes in semen parameters, data on the spontaneous pregnancy rate in couples with infertile men were surprising; in particular, the results of two interventional studies on infertile men with VD deficiency showed the increased chances of a spontaneous pregnancy in the population treated with VD and VD plus calcium supplementation [18,28]. Obviously, these are preliminary data, but they represent an important input to deepen this line of research.

Overall, some limits should be taken into account in evaluating the results of our review. Literature on the topic is poor: in most studies, the sample size was small with a wide study design heterogeneity. Moreover, possible confounding factors (other causes of infertility, co-morbidities, habits, age) could make it difficult to correctly interpret the studies. For these reasons, the clinical importance of all our observations should be validated by further research.

## 5. Conclusions

The results of this review support the hypothesis that VD plays a role in male reproduction. Most of the data highlighted a positive effect on semen quality, both in fertile and infertile male populations. Instead, evidence on any relationship between VD and sperm total count as well as morphology was inconsistent; moreover, research on the association between VD status and sex hormone production did not provide strong results. Experimental studies on the evaluation of VD levels and VD supplementation in men with VD deficiency or insufficiency are scarce to-date and it is not possible to draw conclusions; however, the preliminary data seem promising. This aspect might encourage a new point of view and a new perspective for infertile men and couples in their pregnancy search path. Further dedicated experimental studies are required to clarify the still controversial aspects of this topic.

## Figures and Tables

**Figure 1 nutrients-14-03278-f001:**
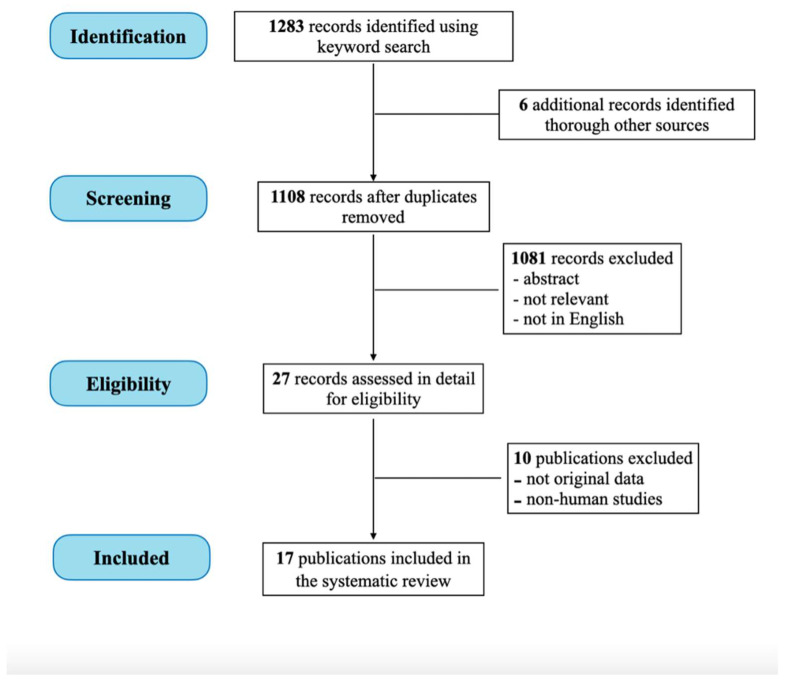
Flow chart of the included studies.

**Table 1 nutrients-14-03278-t001:** Levels of evidence of the articles included in the study research.

Author	Level of Evidence	Study Design	Details (Author’s Definition)
Blomberg Jensen [10]	2b	cohort study	prospective, cross-sectional, observational study
Blomberg Jensen [12]	2b	cohort study	prospective, cross-sectional, observational study
Rehman [13]	4	case-series	prospective, cross-sectional, observational study
Blomberg Jensen [14]	2b	cohort study	prospective, cross-sectional, observational study
Hammoud [15]	2b	cohort study	prospective, cross-sectional, observational study
Tartagni [16]	4	case-series	prospective, cross-sectional, observational study
Blomberg Jensen [18]	1b	RCT	prospective randomized controlled clinical trial
Blomberg Jensen [29]	2b	cohort study	prospective, cross-sectional, observational study
Ramlau-Hansen [30]	2b	cohort study	prospective, cross-sectional, observational study
Yang [31]	2b	cohort study	prospective, cross-sectional, observational study
Waud [22]	2b	cohort study	prospective, cross-sectional, observational study
Abbasihormozi [23]	2b	cohort study	prospective, cross-sectional, observational study
Tirabassi [24]	4	case-series	prospective, cross-sectional, observational study
Zhu [25]	3b	case-control	retrospective observational study
Akhavizadegan [26]	3b	case-control	retrospective observational study
Jueraitetibaike [27]	2b	cohort study	prospective, cross-sectional, observational study
Wadhwa [28]	2b	cohort study	prospective, observational study

**Table 2 nutrients-14-03278-t002:** Main results on correlation between VD serum level with semen parameters and male sex steroid hormone production.

References	Patients (*n*)	Main End-Points	Methods	Sperm Parameters (TC-MT-MR)	T	FSH-LH	Results
Blomberg Jensen [29]	NS (300)	Correlation VD and SQ Correlation VD and SH	− Fresh semen analysis/serum VD − FSH level	TC → MT ↑ MR ↑	NV	FSH →	Significant positive association (MT *p* < 0.05) No significance
Ramlau-Hansen [30]	NS (307)	Correlation VD and SQ Correlation VD and SH	− Fresh semen analysis/serum VD − T and FSH/LH levels	TC ↓ MR ↓	→	FSH → LH →	No significant association No correlation
Hammoud [15]	NS (127)	Correlation VD and SQ Correlation VD and SH	− Fresh semen analysis/serum VD − T and FSH/LH levels	TC ↑ MT ↑ MR →	→	FSH → LH →	Significant positive association (TC,MT *p* < 0.05) No correlation
Yang [31]	NS (195); AS (314) *	Correlation VD and SQ Correlation VD and SH	− Fresh semen analysis/serum VD − T level	MT ↑ MR ↑	→	NV	Positive association No correlation
Tartagni [16]	NS (90)	Correlation VD and SQ Correlation VD and PR/DR	− Fresh semen analysis/serum VD − PR and DR/patient	TC → MT → MR →	NV	NV	No correlation Significant positive association (PR *p* < 0.05, DR *p* < 0.02)
Abbasihormozi [23]	NS (186); AS (92)	Correlation VD and SQ Correlation VD and SH	− Fresh semen analysis/serum VD − T and FSH/LH levels	TC → MT ↑ MR →	→	FSH → LH →	Significant positive association (MT *p* < 0.05) No correlation
Zhu [25]	NS (79); AS (186)	Comparation VD and SQ	− Fresh semen analysis/serum VD	TC ↑ MT ↑	NV	NV	Significant positive association (TC, MT *p* < 0.05)
Blomberg Jensen [14]	AS (1189)	Correlation VD and SQ Correlation VD and SH	− Fresh semen analysis/serum VD − T and FSH/LH levels	TC → MT ↑ MR →	→	FSH → LH →	Significant positive association (MT *p* < 0.05) No correlation
Tirabassi [24]	AS (104)	Correlation VD and SQ Correlation VD and SH	− Fresh semen analysis/serum VD − T and FSH/LH levels	TC → MT ↑	→	FSH → LH →	Significant positive association (MT *p* < 0.001) No correlation
Akhavizadegan [26]	NS (116); AS (114)	Correlation VD and SQ	− Fresh semen analysis/serum VD	TC ↑ MT ↑ MR ↑	NV	NV	Significant positive association (TC,MT,MR *p* < 0.001)
Rehman [13]	NS (186); AS (127)	Correlation VD and SQ Correlation VD and SH	− Fresh semen analysis/serum VD − T and FSH/LH levels	TC ↑ MT ↑ MR ↑	↑	FSH ↓ LH ↓	Significant positive association (TC,MT,MR *p* < 0.001) Significant positive association (T,LH *p* < 0.001)

* Oligo-astheno-teratospermic (OATN). NS: normal semen; AS: abnormal semen; VD: vitamin D; SQ: semen quality; SH: sexual hormones; TC: total count; MT: total motility; MR: morphology rate; T: testosterone; FSH: follicle-stimulating hormone; LH: luteinizing hormone; NV: not valuable. → no correlation; ↑ positive correlation; ↓ negative correlation.

## Data Availability

Not applicable.

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
