# Peer review of "Vitamin D and Male Reproduction: Updated Evidence Based on Literature Review"

_nutrients, 2022, doi:10.3390/nu14163278_

Round 1

Reviewer 1 Report

Dear Authors:

This manuscript has been read before.

I find changes too many.

I think this is a good review article.

Author Response

Dear Prof. Leslie Duan,

We received the comments regarding the Manuscript " Vitamin D and male reproduction. Updated evidence based on literature review " (nutrients-1834508), and we are grateful for your insightful suggestions and the opportunity to clarify our work.

According to the Reviewer 1’s suggestion:

Your manuscript title is “Vitamin D and male reproduction. Updated evidence based on literature review.” Your manuscript has 6% PlagScan. I find one article has the same title “Vitamin D and Male Fertility: An Updated Review”, but they are all different.

Review manuscripts should comprise the front matter, literature review sections and the back matter. The template file can also be used to prepare the front and back matter of your review manuscript. It is not necessary to follow the remaining structure. Structured reviews and meta-analyses should use the same structure as research articles and ensure they conform to the PRISMA guidelines.     

- We are grateful for the positive evaluation of our work. We provided to modify the text as suggested.

Introduction:

[10]. In particular, VDR and VD-metabolizing enzymes were con-comitantly expressed in round and elongated spermatids, vesicles (within the caput epi- didymis), glandular epithelium of cauda epididymis, seminal vesicles and prostate, sug-gesting that VD could be implicated in spermatogenesis and maturation of human sper-matozoa [10–12]. Moreover, VD has been hypothesized to be an important factorfor trans-epithelial calcium transfer in the epididymis. Re-writing this full sentence.

- Thank you for your suggestion. We rephrased this section accordingly.

Results

3.1. VD molecular mechanism

“Successively,………………………..”

Need change.

- Thank you for your criticism. We provided to modify the text as suggested.

Reviewer 2 Report

The work entitled “Vitamin D and male reproduction. Updated evidence based on literature review” reports on the effect of Vitamin D (VD) on male infertility. Data has been compiled on the molecular mechanism underlying VD effects on semen quality, the relationship between VD levels and semen quality, and the effect of VD supplementation on semen quality. Most effects observed were deemed positive, as concluded by the authors. The work is very direct and precise. The information compiled was pertinent but there is lacking more recent work. Were there any information for 2021 and 2022 studies? They are lacking - if there are any they should be studied and included (thus the minor revision for this work). Regardless, this is a very important topic and the work is scientifically sound and of importance. It also allowed the authors to raise important questions that are still in need of answer and that may motivate future investigations.   

Author Response

Dear Prof. Leslie Duan,

We received the comments regarding the Manuscript " Vitamin D and male reproduction. Updated evidence based on literature review " (nutrients-1834508), and we are grateful for your insightful suggestions and the opportunity to clarify our work.

According to the Reviewer 2’s suggestion:

The work entitled “Vitamin D and male reproduction. Updated evidence based on literature review” reports on the effect of Vitamin D (VD) on male infertility. Data has been compiled on the molecular mechanism underlying VD effects on semen quality, the relationship between VD levels and semen quality, and the effect of VD supplementation on semen quality. Most effects observed were deemed positive, as concluded by the authors. The work is very direct and precise.

- We are grateful for the interesting feedbacks about our work.

The information compiled was pertinent but there is lacking more recent work. Were there any information for 2021 and 2022 studies? They are lacking - if there are any they should be studied and included (thus the minor revision for this work). Regardless, this is a very important topic and the work is scientifically sound and of importance. It also allowed the authors to raise important questions that are still in need of answer and that may motivate future investigations.  

- Thank you for the opportunity to clarify this point. We modified the text accordingly and added the following references of 2021 and 2022.

  1. Maghsoumi-Norouzabad, L.; Labibzadeh, M.; Zare Javid, A.; Ahmad Hosseini, S.; Abbas Kaydani, G.; Dastoorpur, M. The association of vitamin D, semen parameters, and reproductive hormones with male infertility: A cross-sectional study. Int. J. Reprod. Biomed. 2022, 331–338, doi: 10.18502/ijrm.v20i4.10905.
  2. Banks, N.; Sun, F.; Krawetz, S.A.; Coward, R.M.; Masson, P.; Smith, J.F.; Trussell, J.C.; Santoro, N.; Zhang, H.; Steiner, A.Z. Male vitamin D status and male factor infertility. Fertil. Steril. 2021, 116, 973–979, doi: 10.1016/j.fertnstert.2021.06.035.
  3. Homayouni-Meymandi, M.; Sotoodehnejadnematalahi, F.; Nasr-Esfahani, M.H. Relationship between Serum Vitamin D in Male, Sperm Function and Clinical Outcomes in Infertile Men Candidate for ICSI: A Cohort Study. Int. J. Fertil. Steril. 2022, 16, 115–121, doi: 10.22074/IJFS.2021.522049.1067.
  4. Kumari, S.; Singh, K.; Kumari, S.; Nishat, H.; Tiwary, B. Association of Vitamin D and Reproductive Hormones With Semen Parameters in Infertile Men. Cureus 2021, doi: 10.7759/cureus.14511.

Once again, we thank the Reviewers for the precious suggestions and the Editors for giving us an opportunity to clarify the issues. We hope that you will appreciate our work. We remain at your disposal for any further detail you might want to discuss.

On behalf of the co-Authors,

Antonio Schiattarella, M.D.

Unit of Gynecology and Obstetrics

Department of Woman, Child and General and Specialized Surgery

University of Campania “Luigi Vanvitelli”

Largo Madonna delle Grazie, 1, 80138, Naples, Italy

Email: antonio.schiattarella@unicampania.it ; aschiattarella@gmail.com

Mobile Phone: (+39) 3921653275